# Time, Memoria, Creation: Receptions of Augustinism in the Philosophical Theology

Tatiana Litvin 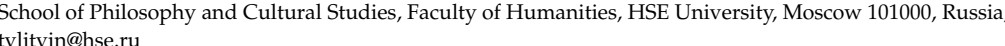

School of Philosophy and Cultural Studies, Faculty of Humanities, HSE University, Moscow 101000, Russia; tvlitvin@hse.ru

**Abstract:** The aim of the paper is to develop a thesis about the potential of phenomenology as a method for analyzing classical ancient texts. The article outlines the key issues of the doctrine of the time of Augustine and raises the question of the principles of phenomenological interpretation. Proceeding from the presentism approach, parallels are drawn between the philosophy of Augustine and the phenomenology of E. Husserl, the issue of duration and structure of the present is especially considered. The philosophy of Augustine includes both theological and epistemological conclusions.

**Keywords:** Augustine; philosophy of time; Edmund Husserl; memory; phenomenology

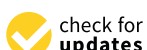



## 1. Introduction

Twentieth-century commentators on Augustine frequently come to the conclusion that understanding the problem of time and memory is not simply related to thinking about creation, and that those two areas of research are not conceivable without each other. Augustine's ontology of time and creation and the problems of measurement of the aesthetic perception of time all reveal an epistemological and structural relationship with the mechanisms of memory. Accordingly, like most aspects of his epistemology, the doctrine of memory, to which the tenth and eleventh book of *Confessions* is dedicated, bears in itself an interpretation of Platonism (and Neoplatonism), which nevertheless acquires Christian forms. In itself, the problem of memory cannot be overemphasized: at the heart of any historicism, be it the poetization of individual experience or the sociology of national identity, lies some idea of the connectedness of events, which in anthropological models can be conditionally called memory.

The fact that Augustine paid much attention to memory is undoubtedly relevant in modern interdisciplinary approaches in philosophical anthropology, including the latest phenomenological studies. After the 'theological' turn in French phenomenology, Augustine's anthropology and epistemology is vividly developed by Jean-Luc Marion, and also in Orthodox theology, by John P. Manoussakis. The theology of creation is also a focus for researchers in natural theology and the philosophy of consciousness, among them Eugene T. Long, Calvin L. Troup, and Roland J. Teske. The historical and philosophical reconstruction of Augustinianism continues to be relevant to modern philosophical anthropology and the philosophy of time, but the question of the theology of creation for the newest philosophical theology remains open. It is impossible to equalize the phenomenological reflection and Augustine's intuition, such as to ignore the similarity of both Neoplatonic and phenomenological approaches to the nature of introspection. To clarify, this similarity is very heuristic for the intuitivism in the philosophy of time.

The aim of this paper is to study the historical and philosophical relationship between time and memory in the context of Augustine's creationism, and to interpret this doctrine phenomenologically. Such phenomenological interpretation must clarify the following question—does the hermeneutics of creation have a dimension in the time experience?

## 2. The Concept of Time According to St. Augustine and Its Phenomenological Interpretation: *Distentio Animi*

The eleventh book of *Confessions* by St. Augustine is usually considered to be the source of all classic questions in the philosophy of time. In polemics with the time doctrines of ancient philosophers, mostly Neoplatonists, Augustine formulated some methodological principles of the philosophy of time, which were later reproduced in their secularized form by phenomenology. I consider three points of his anthropological doctrine of time, namely *distentio animi*; the question of time and memory; and the role of perception and imagination in the process of time constitution. The hypothesis of the research is the following: there are not only historical parallels between Augustine and Husserl as between two temporal philosophers, but there is also a methodological similarity between their approaches. The result of such supposition will be to apply a phenomenological analysis of Augustine's doctrine of time, with an emphasis on its concrete aspects. To justify our method, it is necessary to analyze the two topics of doctrine—the idea of *distentio animi*, and considerations about perception of time, including the aesthetic experience.

Augustine reflects on the relationship between time and eternity, largely relying on the concept of time that Plotinus proposed. Plotinus questioned the relationship between time and eternity in the seventh tractate of the *Third Ennead* (Ennead III, 7), which contains an interpretation of Plato's story about the origin of the world, as expounded in *Timaeus* (Timaeus 28b). In that tractate Plotinus builds his reasoning, starting from the story of Plato in *Timaeus* about the beginning and origin of the world. Already in Plato we see the juxtaposition of eternity and time as a kind of sample (παράδειγμα) and copy, a prototype and image. All created things are created according to the eternal pattern, or rather, the pattern of eternity, to which the Demiurge 'looked', creating the cosmos (Timaeus 28b). Since all created things exist in time, then the 'sky' was created along with time, time (and any created) is also an image. According to Plato, time is a 'moving semblance of eternity', and all its forms, arising and disappearing, past and future, are 'parts' of time—which 'imitates eternity and circles through the number (Timaeus 38a). Moreover, time is born simultaneously with the 'sky', so that it will disappear: 'for the model exists for all eternity, while the universe was and is and always will be for all time' (Waterfield 2008, p. 28, Timaeus 38c)[1]. The creation of planets, stars and all luminaries in general, according to Plato, was necessary to realize the concept of the birth of time. The planets, each equipped with its own movement, participate in the structuration of time, just like the whole geometry of the universe that Plato presented in detail.

Apparently, the correlation between the pre-image and the image is revealed through the number of movements: time functions as a moving image which, however, does not wholly explain the very transformation of the prototype into an image[2]—in other words, the 'secret' of the birth of time. In spite of the fact that the whole set of movements of the planets, their rhythms and velocities, actually forms the world harmony, close to its eternal prototype, and also creates the schemes of the action of all material objects, the question of the origin of time from eternity remains open.

Plotinus also cites that problem of the relationship between the prototype and the image, choosing this difficulty as the central point of reasoning. Werner Beierwaltes, a present-day commentator of Plotinus, indicates that all Plotinus's reflections on eternity and time are determined by the dialectic of prototype and image, which is the question of the origin of time (Beierwaltes 1967, pp. 10–11). Time, according to the Third Ennead, arose because of some activity of the Soul, which 'always wanted to apply to everything else what was known to her in the noumenal world'[3] (Enneads III, 7.11). Time is the life of the sensory world in imitation of the supersensible, and it arises when the activity of the Soul creates events and their sequence, which is time. Namely, 'time is the life of the Soul, which in the process of movement passes from one manifestation of life to another[4]' (Enneads III, 7.11).

Commenting on the time doctrine of Plotinus, it is noteworthy that the influence of the Stoics, Aristotle, and the Epicureans is traceable in his thinking, if through the

prism of Plato's cosmology and mathematics. Plotinus apparently inherits from previous thinkers the doctrine of motion, and the concept of number, as well as measures. Therefore, his consideration of the nature of time can be termed the final one for the entire ancient tradition. However, the so-called psychologization which arises in connection with the concept of the World Soul, belongs to Plotinus himself, and precisely because of this idea the doctrine of time then becomes significant for Christianity. Psychologization—bringing into the cosmological scheme the properties of the 'life' of the soul—emerged, as can be seen from the course of reflection, in the process of simultaneously solving two problems: firstly, the problem of the origin of time, and secondly, the problem of duration as its property. In *Timaeus* and *Physics*, these problems are posed separately. Plotinus, summing up the tradition, combines them in the question of the possibility of measuring time in relation to eternity. In a sense, psychologization further generalizes the formulation of the question of time, in an attempt to answer all questions at once. In any case, we can talk about psychologization as what leads simultaneously to a kind of 'decosmologization',[5] that is, a departure from Platonic creationism.

> Thus, the causal relation between eternity and time in Plotinus correlates with those elements of his philosophy which distinguish it from Plato's, in particular with the hypostases of the One, the Nous, and World Soul. St. Augustine accepts the Neoplatonist idea that time exists in the 'Soul' and is part of the spiritual order; however, due to inconsistency of the idea of World Soul within the Christian doctrine, he shifts the emphasis into the sphere of the individual human soul. The question of eternity is considered in the context of the concept of Creation, although St. Augustine accepts Plotinus's argument about the impossibility of attributing to eternity any temporal characteristics. Time is not part of eternity, not some segment on a 'line' of eternity: it is qualitative, another dimension.
>
> (Confessions, XI, 10.12)

Drawing a distinction between eternity and time, St. Augustine examines time based on the everyday experience. Following his method of considering a subject of reflection as a part of inner experience, St. Augustine describes the daily sensations associated with the experience of time, primarily its duration (Confessions XI, 27.34–35). When he tries to understand what gives us the sensation of duration, the sensation of temporal intervals, which can be short or long, St. Augustine considers the three attributes that we normally ascribe to time as its essential predicates—the past, the present, and the future. As a result, St. Augustine arrives at an idea that will later form the basis of the phenomenological model of consciousness of time: namely, through the present we obtain knowledge about the past and the future. In the present we can exactly differentiate the actual sensation from the memory images of the past, which do not exist *already*, and anticipate events in the future, which do not exist *yet*. Only the present has ontological status, while knowledge about the past and the future derives from our perception of the present. Bringing all three time-predicates to a common denominator, Augustine also points to the subjectivity of time.

In the twentieth century, this idea of present is analyzed in Husserl's phenomenology.[6] The key mechanism of time-consciousness is the constitution of the present.[7] Transcendental Ego as the center of intentional functions is founded on the *actuality* of perception. In the consciousness of immanent objects, Husserl distinguishes not only the properties of objects themselves, but all the ways in which the differences between the properties are understood. Realization of the differences always occurs as if from the perspective of a 'Now-Point' (Jetzt-Punkt) (Husserl [1928] 1985, pp. 390–93), originating from the duration of perceiving the actual Now. Actual perception constitutes the present; it constitutes the temporal[8] objects with their structural definitions, which is the phenomenon of temporal constitutive consciousness. The present is understood as the *way of being* of the transcendental Ego, that is, as a constitutive core and the primary function of the *transcendental I*.[9]

St. Augustine calls time the 'extendedness of mind (soul) (Augustine 1955, p. 203).' (*distentio animi*[10]):

> ' . . . since it is possible that if a shorter verse is pronounced slowly, it may take up more time than a longer one if it is pronounced hurriedly. The same would hold for a stanza, or a foot, or a syllable. From this it appears to me that time is nothing other than extendedness; but extendedness of what I do not know. This is a marvel to me. The extendedness may be of the mind itself'.

> (Conf. XI. 26.33; Augustine 1955, p. 203)

This famous idea allows him to give a new dimension to the problem of duration, and hence to the measurability of inner time. To 'grasp' Now is the way to keep it in mind. If time is measured as a memory of itself—in other words, as a special kind of image in memory—then the very consciousness of time is nothing other than an effort to memorize and to hold in mind.

> 'The span of my action is divided between my memory, which contains what I have repeated, and my expectation, which contains what I am about to repeat. Yet my attention is continually present with me, and through it what was future is carried over so that it becomes past. The more this is done and repeated, the more the memory is enlarged—and expectation is shortened—until the whole expectation is exhausted. Then the whole action is ended and passed into memory'.

> (Conf. XI. 28.38; Augustine 1955, p. 205)

This effort, or as St. Augustine wrote, 'the life of this action' (*vita hujus actionis*[11]) (Conf. XI. 28.38) is the actual present: the effort of consciousness, the attention is focused on the present, through which the future is 'carried over' (*trajiciatur*) and becomes past (Conf. XI. 28.38).

Thus, duration is not a property of the past, present or future, it is rather a property of focusing attention, focusing consciousness. Attention (*attentio*), focused on the present, diffused between memory (*memoria*) and expectation (*expectatio*), this is the effort of consciousness, the 'movement of the soul,' which creates the perception of time. *Attentio, memoria, expectatio* are the three aspects[12] of the act of perception, which correspond to the present, past and future. This three-fold perception and the associated experience of time exist in the dynamics underlying the inner historicity and its external implications—the idea of social history, historicism as a political or ecclesiological scheme.

### 3. Time and Memory: To Remember the Present and to Forget the 'I'

St. Augustine interweaves the question of memory in the problem of past and duration. The memory, or more precisely, the inner memory (*memoria interior*), is the foundation of thinking in general (Conf. X. 8.12–14; Kaiser 1969, S. 23; Teske 2001, pp. 148–58; Troup 1999, pp. 86–87). Impressions and information about the external world enter the memory and remain there in the form of images or concepts about the objects. Mental states and experiences are stored in memory in the form of recollections, the nature of which corresponds to the nature of memory.

> 'I remember the health or sickness of the body. And when these objects were present, my memory received images from them so that they remain present in order for me to see them and reflect upon them in my mind, if I choose to remember them in their absence. If, therefore, forgetfulness is retained in the memory through its image and not through itself, then this means that it itself was once present, so that its image might have been imprinted. But when it was present, how did it write its image on the memory, since forgetfulness, by its presence, blots out even what it finds already written there; (Confessions, X, 16. 25; Augustine 1955, p. 164). Also, memory allows one to remember the existence of God.

> (Confessions X, 24–26. 35–37)[13]

The point is that any synthesis, as such, is built on the ability of memory to constitute the duration (Kaiser 1969, pp. 23–24). Memory is the basis for the interrelation of the experiences of time and continuity; memory 'preserves' the coherence of any content, including those that we conditionally call the connection between the past, the present, and the future. Actually, memory is itself that connectivity, the ability not only to perceive objects of the sensory world integrally, but also to 'assimilate' ideas. The spiritual connection between things assumes both the possibility of their maximum discrimination and detailing, as well as the possibility of changing the contents (Kaiser 1969, pp. 32–35; Troup 1999, pp. 88–90).

This notion of the sense of memory directly clarifies the problem of the present in Augustine's philosophy of time. The present is that which has already fallen into the memory, is 'held' within it, fixed as existing. What Augustine calls the present '*distentio animi*'[14] not only emphasizes the spirituality of the time-image in consciousness (that is, time cannot yet be reduced to sensation), but also points to the complexity of the aporia of memories. Recollection preserves continuity, actualizing experiences, 'leading' them to awareness here and now. However, remembering is a movement, not a rest, continuity includes variability as a repetition of actualization (Kaiser 1969, pp. 47–48). Duration will never be a mechanical reproduction, it is always a living process of 'restoring' the experience.

The meditative character of St. Augustine's reflection is emphasized through discerning and metaphorical language, and it continues the tradition of inner contemplation (*intus cernimus*). The meaning of image and inner imagery goes back to the notion of man's creation in the 'image of God' (Genesis 1:27), that is why imagery as a source of knowledge is the natural foundation of spirituality. St. Augustine also links the notion of memory as an inner space to the topic of the 'inner word' (*verbum interius, verbum cordis, verbum quod intus lucet*). Thus, the way to keep in mind the moment, the remembering, is a special kind of memory, similar to an act of reproduction. Like the other parts of Augustine's epistemology, the doctrine on memory is borrowed from Plotinus,[15] the kinds and functions of memory belong to a capacity of the soul, as the active psychic capacity.

From a phenomenological viewpoint, Augustine writes about objectifying the act of consciousness, which includes different types of reproduction and constitution. The structure of inner perception that constitutes the foundation of time consciousness consists of continuous combinations of elements reflected by subjectivity and passive synthetic unities which connect both objects' properties and differences in a sequence of apprehensions (*Auffassungsakte*). The structure of inner empirical perception is founded thereby on a certain kind of synthesis which allows one to speak about modification not only in the sense of changing, but also in terms of an 'interflow' from one phase to another. The interference of phases is evident in the 'everyday' experience, while in the new experience (such as in social communication) the images of past are 'mixing' with expectations and phantasms. The synthesis is passive but can be structured through reflection.

Jean-Luc Marion analyses the narrative of the *Confessions* and the question of reflection. Marion indicates the significance of the story in the 'first person', when the questions—who is? and who says? (Marion 2008, p. 30)—lead to a conclusion about the speaker's special perspective. According to Marion, Augustine tells the story not strictly 'from the first person', and as though 'instead of himself', locating himself in the present, in the place of himself in the past. This stylistic method of 'imaginary self' typifies Augustine in general (Marion 2008, p. 21) and is crucial for understanding his philosophy. Moreover, that exact method reflects the theological meaning of confession as a form of speech, which contains a narrative and prayer at one and the same time. Prayer creates a very special image of 'I', thus, a special subject which exists in this kind of duality, 'instead of' himself, in the paradoxes 'from the first person'. Memory is something that solves the problem of the paradoxes of 'I' but solves it paradoxically; it does not eliminate the contradiction but completely manifests the impossibility of understanding for subject of himself, the inability of Ego for *cogitatio sui*. Memory gives the unity for consciousness through temporalization (See Manoussakis 2014, p. 19), but has no 'place' in the history of the subject—'the place

of all the thoughts that are not in the world'[16]. According to Marion, the memory as anamnesis olves the problem of consciousness of the subject, in the 'protection him from himself' (Marion 2008, p. 114). Moreover, as the definition of the boundaries of oblivion, as the fullness of actuality of the memory, not only is the absent present (visible), but also keeps the absence absent (invisible), granting both dimensions completeness. In other words, memory is not only the memories of the past, it is also the definiteness of the past, the distance of the past from the subject.

### 4. Imagination and Time: The Creation of the History?

What is the connection between cosmology and prayer? How does experience become the experience of inner history and the perception of time—the experience of fullness of being? From a Christian point of view, this question refers to the special eventfulness of the liturgy that creates the inner spiritual world, and its repeatability sanctifies the inner dialogue of the soul. From a phenomenological point of view, the relationship between prayer and cosmology derives from awareness to experiences of faith. Therefore, the third point of this topic is the aesthetics of Augustine, his theory of music, which does not contradict Husserl's phenomenology, where a musical tone served as a model of a temporal object. The phenomenology of sound can be considered as possessing psychological and aesthetical dimensions, but the wholeness of religious experience of the Eucharist is not possible without the inner historicism of creation and eschatology, and without the repetition of that historicism in each liturgical event. Following Marion's idea that Augustine was the first phenomenologist, we turn to the early period of his work and conversion, which shaped both the aesthetic and cosmological views of the Bishop of Hippo.

Augustine was evidently drawn to music. His early writings in Milan, before his ordination, show that he was overwhelmed by the music he heard in church, because of St. Ambrose who had instituted antiphonal singing and wrote hymns. Ambrose's authority was so high that his style is still a compelling one in the history of church music. Milanese chant took shape distinct from Gregorian chant. It is the oldest style in the western repertory of liturgical music, influenced neighboring regions and spread to central and southern Italy, and as far as central Europe.[17] Augustine introduced several Ambrosian hymns—*Deus Creator omnium, Eterne rerum conditor, Jam surgit hora tertia,* and *Veni Redemptor gentium* (Ibid., p. 695). He was also interested in psalm and hymn singing, and his commentaries on psalms (*Enarrationes in psalmos*) provide some information on the topic. His thoughts on many aspects of music and liturgy are found in other treatises—*Confessionum, Retractationum libri II, De ordine libri II,* and *De Doctrina Christiana libri IV.* The most famous and profound analysis of music theory and its philosophical reflection appears in *De Musica libri VI.* The so-called 'Augustinian canons' would emerge later, in the eleventh–twelfth century, as an organized body with its order characterized by the fully developed liturgical life.

Augustine had a powerful influence on the history and philosophy of music. His ideas were key notions in the transition from the outlook of the Ancient world to that of the Christian world, and aesthetics was formed on the crossroads of Hellenistic directions. Early Christianity's music was connected with theology, and the aestheticization of liturgy was formed as the realization of cosmological order. This idea exists in the later ecclesiology and continues until now in the Orthodox and Catholic liturgy, where the succession of hymns and actions not only glorifies but also reconstructs the history of Creation and Incarnation, with the Eucharist as an emphasis. However, during Late Antiquity, when the dogma was taking shape, the liturgical forms of creationism were a part of that polemics and included elements of ancient philosophical ideas. From the viewpoint of the history of music, the general features of church music are, firstly, improvisation, which differed fundamentally from synagogue worship and, secondly, the primacy of the word (See The New Grove Dictionary 1980, Vol. IV, p. 364). The word as logos 'created' the aesthetical space. The music of the fourth century developed under Asian, particularly Syrian influence, and it is

impossible to affirm that the musical philosophy of early Christianity was based solely on Greek musical theory (Ibid., p. 365).

Concerning cosmology, Augustine's cosmology was based on his various commentaries on Genesis, like other Church Fathers. At the same time he was writing the treatise *De Musica*, from 387 to 391, he was also writing the treatise *De Genesi contra Manichaeos*, from 388 to 389/390. One can hypothesize that the aesthetical realization of cosmology and, perhaps also Augustine's later philosophy of time, developed under the influence of his musical practice.

Thus, the sixth book of *De Musica*, where mostly philosophical theory is concentrated, includes the doctrine on numbers, borrowed from Plato. Plato's *Timaeus* represents the doctrine of creating numbers, which are key elements of cosmogony. It seems that the idea of *distentio animi* and the purpose of measuring time are directly related to the concept of numbers. For St. Augustine, numbers constitute a universal law, a global regularity, around which all life and the spiritual world are organized. In the sixth book, St. Augustine reflects on the types of numbers, which create the foundation for perception. All four types of number manifest different experiences of sensory perception, and so the perception of time is a part of sensibility. As constituting an act of perception, numbers as form are related to the actuality of act, resembling the different modes of temporality—length and brevity (*sonantes* and *occursores*), rhythm (*progressores*), and memory (*recordabiles*) (De Musica, VI). Having determined time as the 'distension of the soul', St. Augustine does not thereby provide an opportunity to claim that time is given only in the senses. If we consider sensuality or perception in general as abilities of the soul, then sensuality itself is manifested by means of time, or in other words, it is represented in a temporal form. The perception is complemented by imagination as formation ability. Moreover, one of imagination's functions is to shape human sensations. Three different components must arrive at unity: the type (*species*) of some objects which can be seen; its image (*imago*), imprinted in the senses, an imaginatory sensation (*sensus formatus*), and the will of the soul (*voluntas animi*), which focuses the senses on the perceived object (*De Trinitate* XI, 2).

Thus, delving into the specifics of the function of imagination, we come to the notion of sensation, which returns us to the issue of duration—thus, the boundary between the past and the present is really hard to measure precisely, because the sensations are formed by the imagination.

This difference between perception and imagination is a typical difficulty in the phenomenology of time-consciousness. Husserl analyzed the notion of imagination as an image-consciousness, in the context of considerations of inner perception (Husserl 1980, pp. 15–18, 35–36). Image consciousness (*Bildbewusstsein*) is similar to the consciousness of time (*Zeitbewusstsein*) in a structural way. A 'fusion' and similarity in appearance of images, structurally replicates the 'fusion' of temporal objects, demonstrating in the description the internal passive mechanism of shaping of temporal form (Husserl 1966, pp. 72–73). The cognition is realized as a coordination or coincidence (*Deckung*) between intention and contemplation (Volonte 1997, pp. 115–19). The imagination allows one to transform the act of perception into an act of consciousness. The system of acts of reproduction and representation (*Vergegenwärtigung*) orders the line of retention-primary impression-protention as reflective levels of self-consciousness and self-cognition (Volonte 1997, pp. 129–34).

## 5. Memory and Creation, Ordo and Experience

As for the problem of creation, one of the meanings of memory is its ontological status as existential, since it clarifies the nature of creation and the variability associated with it. The transition to the ontological plane in a study of Augustine implies that the possibility exists of a kind of equalization of being and knowledge (not thinking), which correspond to each other on the basis of both the Platonic idea (*De Civitate Dei* VIII, 4) and the consequence of the likeness of man to God. Of course, this likeness will never become an identity, but knowledge is eternal, on the grounds that the soul is eternal (*De Trinitate* XV, 15, 24–25),

and knowledge is revealed in the soul. The question of the disclosure of knowledge—in other words, about thinking—demonstrates function of memory exactly. Knowledge and thinking differ only in the degree of actualization, and thinking is precisely actual knowledge, like Plato's 'remembering'. The same applies to knowledge about oneself, that is, the subject also receives knowledge of her 'I' in the form of actualization. In this sense 'I' is always temporal, revealing itself as a temporal being (Kaiser 1969, pp. 116–17). Moreover, if God exists in the strict sense of the word, then a human exists only 'at the moment' of actualizing knowledge about herself. The temporality of a human's existence does not mean the finiteness of life, but on the contrary, the possibility of actualization and of remembering eternal knowledge in the soul, determining our knowledge of ourselves, and memories of the eternal soul.

This same possibility of remembering—actualizing the eternal foundations of our existence—means what we can call the ontology of variability. Creation of the world and human existence is understood, in a certain sense, as *imitatio*. The well-known Christian complexity entailed in understanding *analogia entis*, which was systematically discussed at the time in scholasticism, is also emphasized by Augustine through the indication of contingency (Kaiser 1969, pp. 47–48; Schulte-Klöcker 2000, pp. 35–36), which does not contradict actualization, and at the same time realizes the idea of imitation, similarity. However, since the act of creation cannot be understood as an act arising in time, on the contrary, the question of the beginning is the question of causality, the 'creation' of time acquires the character of aporia even before we can fix any regularities in its nature. Time was created together with the world (according to Plato), with other (Christian) words, eternity was transformed during (at some 'moment') and/or continues to be transformed to date (the continuity of creation). In any event, when asking the question of how eternity (God) creates time, we are either forced to change the register—go to the question of the creation of matter (according to Plato), the numerical patterns, or (in Christianity) the question of the will of God, of his love, because of which the world was created. Augustine, as can be seen from the Confession, takes the second step.

Arguing about time and variability, Augustine takes into account the possibility of a negative term (*Confessions* XII, 38; Schulte-Klöcker 2000, pp. 53–54), that is, the ability to affirm the difference of time from eternity, to understand time as 'non-eternity', respectively, as a characteristic of the being of things. This difference does not, however, become the complete opposite of eternity, because it is set by it, and instead becomes a sort of order which organizes the material world, embodying God's presence in it[18] (*Confessions* I, 10; *Confessions* VII, 21). Thus, variability is the order of control of the world, based on the eternal fullness of God.[19] There is no reason to believe that variability can be reduced to a pattern of sequence (temporal moments); on the contrary, variability based on completeness is the possibility of the unfolding of being, and time as its order plays the role of a transition from one (any) state to another, from the state of non-being to being.

This idea of the variability of abiding eternity seems to be a key one in Augustine's time doctrine. It is that state of time which he defines as *distentio* (*Confessions* XII, 11, 12), 'dispersion', 'stretching', 'scattering of moments', a kind of movement as a whole 'desire for being (and for nothingness)',[20] which is also natural, as a world good, but we can only understand it through reflection and questioning. *Distentio* as the fullness of presence can solve the problem of the present which, like every moment of the created world, involves an aspiring and already-becoming movement towards being.

It should not, however, be forgotten that this fullness does not come from the nature of the movement, but from the nature of the divine presence, and therefore a psychological reading is also necessary. The fact that the *distentio* as an ordo exists in the soul may not be principally ontological (that is, the fullness exists and the world is created in an order, regardless of our understanding), but without this fact the measurement of time is impossible. Even if we imagine that we can dispense with the notions of the past and the future (*Confessions* XI, 37)—which is unlikely in the ontological sense, taking into account not only creationism but also Christian eschatology—even then the universality of the

fullness of being needs to be interpreted. The interpretation is possible only as a sequence, a quantitative statement of the essence. In other words, in language there is already a need to measure time—time as the fullness of being and the presence of God.

Concluding this study with an analysis of modern receptions of the question of creation, we will reveal the key to philosophical theology in which one can find elements of Augustinianism. In the understanding of creationism, one can conditionally distinguish the concept of creation as an object of comprehension in natural theology, and creation as a political category in which the design of 'eternal being', the City of God on Earth, is realized. These two meanings often coincide, since in natural theology the revision of the concept of experience is actualized and, following the nineteenth-century disputes between science and religion, needs a new thematization. Philosophical theology after Karl Barth, Rudolf Bultmann, and Paul Tillich raises questions about new forms of substantiation of experience, assimilating not only the rhetoric of existentialism, but also its theory of cognition, the appeal to 'living' forms of experiences and judgments, and expands the boundaries of both theological and ethical reflection. After Hannah Arendt's dissertation which was dedicated to the concept of love in Augustine, Augustinianism becomes a part of political reflection and remains until now one of the directions of Christian liberalism.

In this regard, in parallel with the Aristotelian–Thomist tradition for natural theology, there is increased urgency of the tradition deriving from Augustine and, thanks to Anselm, Bonaventure, and Duns Scott, representing a fundamentally different one, in comparison with the Aristotelian–Thomist tradition of the proof of the existence of God and of God-knowledge. Its difference is that it includes the issue of free will, which is contained in the first book of the same treatise of Augustine and receives rather detailed reflection in further philosophical theology. Not only the idea of salvation, but in general, the proof of the world order and hierarchy (Mackey 2011, pp. 16–18) proceeds from the experience of direct contemplation given as free action. Augustine considered the existence of such an ability as *sensus interior* which, together with the external (*corporis sensus*) and reason, forms the foundation of understanding and existence. The idea of an inner feeling not only reflects the immediacy of self-reflection and thus develops the neo-platonic epistemology characteristic of all the writings of Augustine. From the historical and philosophical point of view, justified in this way interiority will become the basis of the idea of the autonomy of the subject, the autonomy of consciousness and personality, politically and ontologically significant for the whole of the new European culture (Taylor 1989, pp. 131–32). Accordingly, its involvement in the proof of the existence of God and the hierarchy of order assigns a special place to the natural theology of Augustine (Taylor 1989, pp. 133–34).

Thus, Long, rightly noting that the concept of experience up to the twentieth century was developed in two key traditions—British empiricism and Kant's philosophy—indicates that in modern times this notion should be revised, including from the standpoint of natural theology, whose function is to mediate between 'everyday' and scientific discourse on the one hand, and with theological discourse on the other. Using the idea that experience is what is given in reflection and, accordingly, implies the consideration of experience precisely as suffering (or gift) and the need for reflexivity, not just descriptiveness or articulation, Long suggests the following option for explaining the transcendental dimension of human experience and its boundaries. First of all, experience can be understood as a contingent experience, as an experience that can include the most extraordinary manifestation of miracles and freedom. Contingency need not be understood in a universal sense (and only in the sense of a cosmological argument); rather, it is about the existential freedom of which Sartre wrote, and even more—Jaspers—who understands it as a kind of completeness of self-reality realized in one's existence and that is not unimportant in the freedom to be next to the Other. This position undoubtedly reconciles several questions at once: how is religious experience possible, what is a person, and why does the existence of God remain an integral part of the problem of determining the boundaries of human cognition? In addition, this position points to the role of phenomenology for the interaction of modern approaches in the dialogue of natural theology and science. Experience is



any comprehension that can be carried out both by rational schemes of causality, and in existential or aesthetic experience. Although it can be concluded that the very concept of reflection in this case is either blurred or complicated by a system of phenomenological description, the question about the boundaries of the mind remains unanswered, in other words, about the abilities of thinking, and the cognitive possibilities of understanding the world and oneself.

## 6. Conclusions

The phenomenological interpretation thus enables us to clarify the theory of time posited by St. Augustine, and to analyze its epistemological background. The phenomenological reduction and transcendental goals of this method lead to a specific kind of secular philosophy of self-consciousness, but the comprehensive phenomenological analysis lets us understand the nature of time and human experience. St. Augustine interweaves the question of memory in the problem of time and the function of imagination in the process of self-cognition—this statement of question constitutes the phenomenology of inner time-consciousness in both Husserl's and Augustine's theories. The experience of inner perception of time orders not only the aesthetical experience but also the experience of fullness of being.

**Funding:** This research received no external funding.

**Conflicts of Interest:** The author declares no conflict of interest.

## Notes

1.   In other words, time as a created thing will not exist forever.
2.   For more on the transformation of paradigm into image, f.e. the difference between paradigm and image see also classical analysis of Mesch (2003, pp. 145–57).
3.   See also the translation of Enneads III.7. 11: 'For the Soul contained an unquiet faculty, always desirous of translating elsewhere what it saw in the Authentic Realm, and it could not bear to retain within itself all the dense fullness of its possession.' (Page 1956).
4.   See also: 'Would it, then, be sound to define Time as the Life of the Soul in movement as it passes from one stage of act or experience to another?' MacKenna & Page f. 5.
5.   Beierwaltes uses the term 'Entkosmologisierung', the translation of which as 'decosmologization' seems appropriate, among others because in modern philosophy of religion, the 'demythologization' of Rudolf Bultmann is consonant. Although Beierwaltes himself does not mention him, in my opinion, the transitional role of Neoplatonism in many ways contributed both to liberation from ancient myths and the emergence of a new anthropology.
6.   In the introduction of Vol. 10 of *The Husserliana*, Rudolf Bernet writes that Husserl's phenomenology of time could well be called 'marginal notes' to the *Confessions* of Augustine. See (Husserl 1985, p. XI). See also (Bernet 2009, p. 117). For more on Husserl's phenomenology of time see (Husserl 2001, 2006).
7.   The key research on the question of present is (Held 1963).
8.   Temporary as a continuing object of perception, for example, sound.
9.   For more about the transcendental 'I' and self-temporalization, see (Brand 1955, pp. 67–68, 71–74).
10.   The origin of the term is unclear, but there is a hypothesis that the term may have been borrowed from Basil of Caesarea. See: (Callahan 1958, pp. 437–54).
11.   Basic reference for each latine term: Augustinus Hipponensis. Confessiones. Pl 32. Jacques-Paul Migne, 1845.
12.   Kurt Flasch mentioned that this trinity does not have theological ground. See (Flasch 2004, S. 219).
13.   For more on self-cognition through the idea of remembering God (O'Daly 2001, pp. 41–43).
14.   Modern commentators often wonder whether it is possible to understand *distentio animi* as any extension expressible in spatial terms. This question is fully justified, given that Augustine describes memory in the metaphors of space (Confessions X, 12), but the length is not inherent in the present. Most authors are inclined to this opinion. See (Futch 2002; Humphries 2009, pp. 80–84).
15.   For numerous other parallels between the two thinkers, see (O'Daly 2001, pp. 41–43).
16.   (Marion 2008, p. 113). For a deeper analysis of absence and presence from the psychanalytical perspective see (Manoussakis 2016, pp. 54–55).
17.   For more on the Ambrosian (Milanese) Rite, see (The New Grove Dictionary 1980, Vol. I, p. 314).
18.   We can say that being in time (in the material world) does not prevent us from being with God (in eternity).

[19]     Kaiser, fn. 18, (Kaiser 1969, pp. 47–48, 144–45; Callahan 1967, pp. 24–26, 40–41).

[20]     Humphries expresses the thesis that in understanding *distentio animi* there is no need to distinguish between anima and animus, therefore, the present is equally created by both the soul and the mind. *Distentio animi* is a function of memory in its ability to actualize the semblance of time and eternity, fn. 23, pp. 75–101. His point of view differs from the well-known position of Teske, who believed that *distentio animi* refers exclusively to the Plotinus world soul. See (Teske 2001, p. 32); Cf. (Kaiser 1969, p. 146; Schulte-Klöcker 2000, pp. 69–70).

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
