# Peer review of "Time, Memoria, Creation: Receptions of Augustinism in the Philosophical Theology"

_religions, doi:10.3390/rel13080679_

Round 1

Reviewer 1 Report

Overall, this is a well written paper that deserves publication subject to certain additions and a few clarifications.

I find the statement about the methodological similarity of Augustine and Husserl on lines 32-33very axiomatic. It needs to be explained. Why was Husserl so similar to Augustine? Most of the philosophers of his time had studied theology and classics and hence the author needs to make a valid point about the similarities, beyond simply observing them. In trying to find who else wrote about Husserl and Augustine I bumped onto a 2016 article by Nicholas de Warren, Augustine and Husserl on Time and Memory, Quaestiones Disputatae 7. Now, this work is not included in the biblio but it seems more than probable that the author is the same. I wonder how different is this article to the published work. If this was done to preserve author anonymity, well, it needs to be amended and frankly it has not worked! 

I did say I did not detect plagiarism and have no ethical concerns about the piece because in my view my role is to query the level of similarity between the two articles BUT NOT COMPARE THEM or determine the case. I am a reviewer in this case, not an examiner. But I strongly urge the author and publishers to double-check and make sure that even the slightest suspicion of self-plagiarism is removed.

Line 88 that or what was known?

I wonder what the author would have to say about Lakoff’s later theory of time-is-space (for example, see file:///C:/Users/eanag/Downloads/_journals_fdl_34_1_article-p53-preview.pdf). I think it applies very well to what Augustine was trying to achieve with the concept of distentio animi. This is also especially appropriate for lines 221-222 of the text where the author refers to metaphorical thinking.

Line 272: is not or does not contradict?

Line 318 correct spelling of Timeus = Timaeus. For Augustine and music, consider citing Anagnostou-Laoutides, E. 2021. “Attuning to the Cosmos: the Ethical Man’s Mission from Plato to Petrarch,” in C. Monagle (ed.), The Intellectual Dynamism of the High Middle Ages, A Festschrift in Honour of Professor Constant Mews, Amsterdam: Amsterdam University Press, 249-278.

Again here, I sense that the author must deal not just with Husserlian phenomenology but also with later versions that tried to tackle metaphor as a form of expressing our phenomenological realities. For the philosopher as the best musician, a basic Platonic metaphor, and how Augustine adapted it, see chapter above.

Nn44 and 45 came through with odd fonts.

Author Response

Thank you very much for your opinion. I am agree with you proof-reading (in red), and remarkes about methodology. I added some considerations to clarify the comparison of Husserl and Augustine (lines 35-38). 

Concerning Lakoff’s theory: it is not necessary to unite space and time in the  philosophy (may be in physics it can be useful, but not for philosophy) and to consider time as space metaphor. Of course, it is just my opinion.

Thank you for the suggestion to use ‘’The Intellectual Dynamism of the High Middle Ages’’ (2021). I requested the file and hope use this analysis in my future research.  

P.S. I understand your concerns, I thank you for your attention and for comparing my modest work with the thoughts of a well-known specialist. I am not Nicolas de Warren and have not read this paper unfortunately. Of course I will fix this lacuna. I am absolutely sure that the editors follow all the rules of ethics and the check for plagiarism is carried out before the article is sent to you.

Reviewer 2 Report

The use of phenomenological analysis of medieval texts is not new and has been used especially in the French tradition, not only by Marion or Courtine, but also, for example, by E. Falqué with respect to the work of St Bonaventure. In this sense, the article is not original, but this does not detract from its merit as a phenomenological analysis of an author like St. Augustine.

However, two problems arise. One relates to the methodology itself and the other to the analysis of Augustine's subjectivity by comparing it with that of Husserl.

Regarding the methodological analysis, the phenomenological analysis can be interesting in order to be able to see the aspects that phenomenology wants to see of concepts and positions taken by the authors it analyses. But the result is a phenomenological philosophy that is far from what the author really wanted to say. In short, we can make a phenomenological reading of an author, but the methodology does not help us to really understand that author. We can only see aspects that can be interpreted phenomenologically. This is even more accentuated when an approach is made, not only from phenomenology, but also from Husserl himself.

This brings us to the second problem. Husserl's analysis of Augustine perverts the philosophy of the Bishop of Hippo. The schema proposed by Augustine is extra nos - intra nos - supra nos, where exemplary ideas play a decisive role. The Hussrelian scheme is alien to this external and exemplary gradation because epojé prevents it. We are faced with two incomensurable positions.

What I have just pointed out does not prevent the publication of the article, because the author argues and writes his thought well. Even if I suggest points for discussion and express my disagreement with the approach, this does not prevent it from being really interesting to be published in a way that enriches the philosophical debate.

Author Response

Thank you very much for your opinion. I am agree with you remarks about methodology and added some considerations to clarify the comparison of Husserl and Augustine (lines 35-38)

Round 2

Reviewer 1 Report

I am satisfied by the author's attempt to respond to my concerns.